# Efficacy of Blunt Force Trauma, a Novel Mechanical Cervical Dislocation Device, and a Non-Penetrating Captive Bolt Device for On-Farm Euthanasia of Pre-Weaned Kits, Growers, and Adult Commercial Meat Rabbits

**DOI:** 10.3390/ani7120100

**Published:** 2017-12-15

**Authors:** Jessica L. Walsh, Aaron Percival, Patricia V. Turner

**Affiliations:** Department of Pathobiology, University of Guelph, Guelph, ON N1G 2W1, Canada; jessica.walsh@gov.ab.ca (J.L.W.); percivaa@uoguelph.ca (A.P.)

**Keywords:** cull, rabbit, animal welfare, physical methods, on-farm euthanasia

## Abstract

**Simple Summary:**

Developing effective and humane on-farm euthanasia methods is essential for all livestock industries to ensure that animals do not suffer and are killed humanely. Approved methods are lacking for commercial meat rabbits, potentially leading to poor welfare. We assessed several methods of on-farm killing of cull rabbits of different ages to determine which methods were most effective and humane. These included blunt force trauma (the most commonly used method on rabbit farms), a novel mechanical cervical dislocation device, and a non-penetrating captive bolt device. We evaluated method effectiveness by examining animal reflexes and behaviours after applying the method as well as by examining radiographs of rabbit heads for signs of skull damage, and by assessing the degree of trauma to the brain through dissection and microscopy, because more trauma is generally correlated with enhanced method effectiveness and irreversibility. We found that blunt force trauma resulted in an unacceptably high failure rate, particularly in mature rabbits, whereas the mechanical cervical dislocation and non-penetrating captive bolt devices were both highly effective for killing rabbits humanely and irreversibly. The non-penetrating captive bolt device was the most effective with a 100% success rate and could be used on all rabbits weighing more than 150 g.

**Abstract:**

The commercial meat rabbit industry is without validated on-farm euthanasia methods, potentially resulting in inadequate euthanasia protocols. We evaluated blunt force trauma (BFT), a mechanical cervical dislocation device (MCD), and a non-penetrating captive bolt device (NPCB) for euthanasia of pre-weaned kits, growers, and adult rabbits. Trials were conducted on three commercial meat rabbit farms using 170 cull rabbits. Insensibility was assessed by evaluating absence of brainstem and spinal reflexes, rhythmic breathing, and vocalizations. Survey radiographs on a subsample of rabbits (*n* = 12) confirmed tissue damage prior to gross dissection and microscopic evaluation. All 63 rabbits euthanized by the NPCB device were rendered immediately and irreversibly insensible. The MCD device was effective in 46 of 49 (94%) rabbits. Method failure was highest for BFT with euthanasia failures in 13 of 58 (22%) rabbits. Microscopically, brain sections from rabbits killed with the NPCB device had significantly more damage than those from rabbits killed with BFT (*p* = 0.001). We conclude that BFT is neither consistently humane nor effective as a euthanasia method. MCD is an accurate and reliable euthanasia method generally causing clean dislocation and immediate and irreversible insensibility, and the NPCB device was 100% effective and reliable in rabbits >150 g.

## 1. Introduction

Livestock producers must have effective methods in place to euthanize sick, injured, and cull animals [1,2,3]. Few guidelines are available to assist meat rabbit producers in making on-farm euthanasia decisions, including recommendations as to which methods are most appropriate for different ages and sizes of animals. A recent survey of Canadian commercial meat rabbit producers established that the most common on-farm euthanasia method in use is blunt force trauma (BFT) [4]. BFT was used historically in Ontario abattoirs to stun rabbits for slaughter, but was discontinued in 2005 due to concerns surrounding efficacy and operator fatigue [5]. The American Veterinary Medical Association (AVMA) also encourages alternative methods to BFT because there is a large margin for error and operator fatigue, in addition to observer esthetic issues [6]. One possible alternative euthanasia method is the non-penetrating captive bolt device (NPCB) device, which is designed to deliver lethal brain trauma without penetrating the skin, reducing potential biosecurity and observer esthetic concerns. This device is used routinely for killing meat rabbits in Ontario abattoirs and has been validated for euthanasia of turkeys and piglets, with killing efficacy rates of 96% and 100%, respectively [7,8] For these indications, the device has been applied twice in rapid succession.

Cervical dislocation (CD) is another method commonly used for rabbit euthanasia [4]; however, it has been suggested that CD is inappropriate for rabbits weighing >1 kg due to their large neck muscle mass [6,9]. Various mechanical CD devices have been used in other species, such as poultry and mice [7,10]. To be humane, the device must cause a clean dislocation between the skull and first cervical vertebrae (C1), rupturing major blood vessels, as well as causing a rapid loss of sensibility [7,10]. These devices have not been evaluated in meat rabbits.

Our objective was to evaluate the effectiveness of several physical methods for euthanizing adult and juvenile rabbits, including BFT, a novel mechanical cervical dislocation device (MCD) device and a NPCB device, as determined by onset of rapid and irreversible insensibility and degree of induced brain damage. Based on previous studies examining the efficacy of the NPCB device in poultry, we hypothesized that this device would be most effective and humane for farmed rabbits.

## 2. Materials and Methods

### 2.1. Animals

The procedures and protocol for this research were reviewed and approved by the University of Guelph Animal Care Committee (AUP3366). The University of Guelph holds a Good Animal Practice certificate issued by the Canadian Council on Animal Care and is registered under the Ontario Animals for Research Act. All rabbits used in this study were ones targeted for euthanasia by producers and procedures were conducted on-farm, necessitating convenience sampling. The majority of rabbits that are culled on-farm are young animals (i.e., preweaned kits and growers) with lesser numbers of does and very few mature bucks, given their very small relative population on any given farm. Necessity for euthanasia of individual rabbits in this study was determined by operators and reasons included sick or injured rabbits, unthrifty rabbits that were unlikely to survive to market weight, and rabbits reaching the end of reproductive utility. Three commercial meat rabbit farms (Farm 1 = 500 does, Farm 2 = 150 does, Farm 3 = 600 does) from across southwestern Ontario were recruited to participate in this study. Producers were experienced in rabbit production and had raised meat rabbits commercially full-time for at least eight years. Between May and August 2015, a total of 170 rabbits were euthanized over 28 trial days by the following methods: BFT (*n* = 58), MCD (*n* = 49), and NPCB device (*n* = 63). Minimum sample size requirements to detect differences with an error rate of 5% between techniques and age groups (using 10% variability) was estimated from previous euthanasia studies [5,9] as requiring at least 16–20 rabbits per group. Live weights were collected prior to euthanasia and body length was measured post mortem from the tip of the nose to the tail to calculate body mass index [BMI = weight (kg)/length (m)^2^]. All methods were tested on New Zealand white-like rabbits across three age groups: pre-weaned kits (2 to 5 weeks), growers (6 to 12 weeks), and adults (>12 weeks). 

### 2.2. Euthanasia Techniques

Euthanasia techniques were unbalanced across the three farms, because the purpose of the study was to compare the novel NPCD, which was not in use on any farm, against the two other methods commonly in use on working meat rabbit farms. All three farms (i.e., three different operators) used BFT as a routine euthanasia method. Blunt force trauma was the primary method for all age groups of rabbits on Farms 2 and 3 (both with male operators), and was used on pre-weaned kits only on Farm 1 (a female operator). The BFT method used by producers varied, but in all cases, the rabbit was suspended by its back legs prior to striking the animal on the head with a heavy object or striking the rabbit against a hard surface. 

A novel mechanical cervical dislocation device was the primary method used by Farm 1 for larger post-weaned rabbits. The V-shaped, stainless steel device was wall-mounted at approximately shoulder height (Rabbit Wringer, http://www.rabbitwringer.com, West Grove, PA, USA) (Figure 1). The rabbit’s head slides into the V-shaped wedge and the device acts to secure the rabbit’s head prior to the operator applying a downward force to the hips and back legs. The device allows for alteration of the V-shaped opening according to rabbit size by means of an adjustable neck plate.

The NPCB device (Zephyr-E, Bock Industries, Philipsburg, PA, USA) is a commercially available modified pneumatic nail gun weighing 750 g, with a mushroom-shaped nylon bolt head that extends 2 cm from the end of the barrel (Figure 2). The device attaches to a standard air compressor, set to the desired pressure, which was validated in an initial trial. Although a single discharge appeared to be sufficient for inducing immediate insensibility in all rabbits, the device was discharged twice in rapid succession in the same location based on recommendations in Erasmus et al. [5] and because this is how the device is used in Ontario abattoirs for killing meat rabbits. Two discharges can be made rapidly and without hesitation with this device by depressing the trigger twice. The NPCB device was used exclusively by trained research personnel during this trial (one operator, JLW). When the NPCB device was used, rabbits were gently restrained in a plastic container with non-slip flooring, with the operator’s hand resting on the shoulder blades with thumb and forefinger placed gently around the neck of the rabbit.

### 2.3. NPCB Device Validation Trial

An initial pilot study was conducted at the University of Guelph on 22 donated rabbit cadavers of varying sizes and ages, weighing between 0.9 kg to 4.7 kg, to determine the appropriate placement and required pressure for the NPCB device. The correct location and pressure were determined by dissection of the head following application of the device, looking for maximum brain damage without breaking the skin or injuring other anatomical sites, such as the eye or frontal sinuses, which might contribute to poor esthetic outcome as well as increased biosecurity risk. From this initial cadaver work, and based on preliminary observations and discussions with abattoir personnel who use this device routinely by discharging twice in rapid succession for killing rabbits prior to slaughter, it was determined that the NPCB device should be positioned to rest lightly on the frontal and parietal bones, in the center of the forehead, with the barrel placed in front of the ears and behind the eyes (Figure 2). The required pressure setting for the air compressor was determined through dissection, assessing degree of skull fracture. Kickback from the device was a potential concern for operator safety as well as possibly inhibiting speed of delivery of the second discharge if repositioning is needed. Pressure that was too high resulted in external bleeding secondary to skin trauma, a biosecurity risk and esthetic concern. For this reason, the appropriate pressure was determined based on degree of skull development and thickness rather than applying a maximum pressure to all animals, as is used in abattoirs. The appropriate pressure for sufficient brain trauma and skull fractures was determined to be 621 kPa (90 psi) for adult rabbits (>12 weeks of age), 483 kPa (70 psi) for growers (6 to 12 weeks), and 379 kPa (55 psi) for pre-weaned kits (150 g and larger, ≤5 weeks of age). A minimum pressure of 345 kPa (50 psi) is needed for the NPCB device to discharge.

### 2.4. Assessment of Insensibility and Scoring of Euthanasia Outcome

Cull rabbits were randomly assigned using a random number generator (www.random.org) to the operator to use their preferred euthanasia technique for the size and age of rabbit or to research personnel (JLW) to use the NPCB device. Blind scoring techniques could not be used because it was obvious at the time of examination which method had been applied. Immediately after each euthanasia method was applied, an observer assessed the rabbit for insensibility by evaluating rabbit reflexes, postures, behaviours, and palpable heartbeat. Reflexes assessed included pupillary, palpebral, toe and ear pinch withdrawal, inner nostril poke, and corneal. Reflexes were selected based on use in other studies examining insensibility in rabbits [11,12]. Nostril poke required a pointed tool to be applied to the medial surface of the nares and elicited a variable head withdrawal response in sensible rabbits. Because of this, this test was eliminated from subsequent euthanasia trials. The palpebral reflex was assessed by gently tapping the skin around the eye and was considered present if the rabbit blinked in response. The pupillary reflex was evaluated by shining a white light-emitting disposable penlight (Safe Cross First Aid LTD, Toronto, ON, Canada) into the rabbit’s eye and was considered present if the rabbit tried to avoid the bright light or if the pupil constricted in response. The toe and ear pinch withdrawal reflexes were assessed by gently pinching the fold of skin between the rabbit’s toes or the rabbit’s ear and was considered present if the rabbit moved its foot or head away in response. The corneal reflex was assessed by touching the surface of the cornea and was considered present if the rabbit blinked in response. If the animal was not immediately insensible or it returned to sensibility, as determined by return of rhythmic breathing, vocalizations or reflex responses, the euthanasia technique was deemed a failure, and the same method was immediately reapplied. Rhythmic breathing was defined as regular nostril flaring, differentiating it from gasping. Vocalizations were defined as a high pitch squeal, differentiating them from sporadic low frequency sounds made as air passively escaped from the lungs of some rabbits at death.

Time to death was recorded based on loss of reflexes, lack of movement, and onset of palpable cardiac arrest. The occurrence and duration of involuntary movements were timed and recorded. Involuntary movements were classified as tonic convulsions (rigid extension of limbs) and clonic convulsions (leg paddling). Cardiac arrest was determined when a heartbeat could no longer be palpated or auscultated. For consistency, personnel acting as the observer, the recorder, and the NPCB device operator were constant throughout all trials. Prior to beginning trials, personnel trained and practiced the various measures to judge insensibility and death under veterinary supervision (PVT) at an abattoir that used the NPCB device for rabbit killing.

Rabbit cadavers were taken from farms back to the University of Guelph for gross dissection and assessment of damage. Brain damage assessments were conducted only on rabbits that were successfully killed on a first attempt by any method. 

### 2.5. Survey Radiographs

Survey radiographs were conducted at the OVC Health Sciences Centre on four randomly selected growers from each euthanasia technique, except in the BFT group where one rabbit (5 weeks of age) taken for radiographs was classified as a pre-weaned kit. Both dorsoventral and lateral views were taken and radiographs were qualitatively assessed for location and degree of skull injury as well as subcutaneous hemorrhage.

### 2.6. Macroscopic Assessment of Tissue Damage

Dissections and macroscopic examinations for scoring gross hemorrhage were conducted on all rabbits successfully euthanized on first attempt (154). All macroscopic assessments were conducted by the same two people who trained and scored together, to ensure consistency and a consensus score was recorded. A semi-quantitative gross scoring system was developed based on Casey-Trott et al. [13] and Veltri and Klem [14] and was used to assess the severity of skull fractures, subcutaneous hemorrhage and subdural hemorrhage (Table 1). After removing the scalp, the degree of subcutaneous hemorrhage was scored from the top of eyes to the base of the skull. Similarly, the neck was excised exposing the first four cervical vertebrae to score perispinal hemorrhage. Skull fractures were scored for severity (number of fractures and degree of fracture displacement). Vertebral dislocations and fractures were assessed (present or absent) through palpation and dissection. Following superficial scoring, the skull was opened with a Stryker saw (Mopec. Oak Park, MI, USA), the dura removed, and the amount of hemorrhage on the dorsal and ventral surface of the brain was examined and scored. The total subdural hemorrhage score was calculated as the average of the dorsal and ventral score for each rabbit brain.

### 2.7. Microscopic Brain Hemorrhage Scoring

After macroscopic analysis, all brains were placed in 10% neutral buffered formalin, placed on a shaker for 72 h and stored pending trimming. Thirty-eight brains were selected for microscopic hemorrhage scoring as follows: 11 brains from the growers that were radiographed, three randomly selected brains from pre-weaned kits for each euthanasia method (including the one pre-weaned kit that was radiographed) and six randomly selected brains from adult rabbits for each euthanasia method. Three sections were trimmed from each brain by the same person to represent the hindbrain (cerebellum), midbrain (thalamus), and frontal cortex. Brain sections were embedded in paraffin, sectioned, and stained with hematoxylin and eosin (Animal Health Laboratory, University of Guelph). Sections were randomized and assessed by a veterinary pathologist (PVT) blinded as to euthanasia method and rabbit age. The degree of subdural and parenchymal hemorrhage was scored for each brain section using a scale of 0 to 4 [11] based on the relative area of the brain section affected: (0) 0%, (1) <5%, (2) 5% to 10%, (3) 11% to 30%, (4) >30%. Differences in damage among the three brain regions were assessed by comparing scores from each region. Overall degree of subdural and parenchymal hemorrhage in the brain was determined using the highest score for subdural and parenchymal hemorrhage in any of the three regions from the same rabbit. 

### 2.8. Statistical Analyses

Statistical analyses were performed using SPSS (SPSS Statistics for Windows, Version 23.0. 2014. IBM Corp. Armonk, NY, USA), with *p* < 0.05 accepted for significance. All values are reported as mean ± SE. Data was initially evaluated for normality (Shapiro-Wilk test) and homogeneity of variance (sphericity) and transformed, if not normally distributed. An independent sample *t*-test was used to compare body weight and BMI between the replacement and breeder age groups. A Chi-square test for independence was used to test for associations between method, age, and method failure rate. Cramer’s V test was assessed to determine the strength of the association. Duration of clonic and tonic convulsions were combined for total time convulsing for analysis of involuntary movements. Time convulsing was log transformed and analysed using a one-way ANOVA and post hoc Tukey’s test with treatment as the independent variable. All gross and microscopic hemorrhage scores and skull fractures were rank transformed. Macroscopic scoring of damage was analysed using a Kruskal Wallis one-way ANOVA with method and age group as the independent variables. Microscopic brain hemorrhage scores were analysed using a Kruskal Wallis one-way ANOVA with method, age group, and brain section as independent variables. Data was split by treatment to allow for comparisons within treatment groups and a Bonferroni correction was applied for multiple comparisons.

## 3. Results

### 3.1. Assessment and Scoring of Euthanasia Outcome

Euthanasia methods were evaluated for 170 rabbits across the three age groups. Pre-weaned kits weighed 0.2 kg ± 0.03 with an average BMI of 10.6 ± 2.4, and growers weighed 1.4 kg ± 0.1 with an average BMI of 12.7 ± 0.7. Replacement rabbits did not significantly differ in body weight (3.3 kg ± 0.2) compared to breeding rabbits (3.5 kg ± 0.1), t(40) = −0.69, *p* = 0.50. Similarly there was no significant difference in BMI for replacement rabbits (15.9 ± 2.0) compared to breeding rabbits (16.1 ± 2.1), t(40) = −0.23, *p* = 0.82 and the two age groups were combined for the adult age group. Rabbits in the combined adult age category weighed on average 3.4 kg ± 0.1 with an average BMI of 16.0 ± 0.3.

A euthanasia method was judged to have been effectively applied if the rabbit demonstrated immediate and irreversible insensibility, based on no vocalizations, no response to reflex testing and lack of rhythmic breathing until cardiac arrest. Of 170 rabbits in the study population, 154 or 91% were euthanized successfully (Table 2). A Chi-squared test was performed and a medium strength association (Cramer’s V = 0.33) was found between method used and probability of method failure, χ^2^ (2, *n* = 170) = 18.67, *p* < 0.001. Probability of method failure was highest for BFT (Table 2). BFT data collection was discontinued on Farm 2 after five of eight rabbits failed to achieve insensibility and death after the first application. The highest probability of method failure across all three farms occurred when adult rabbits were euthanized with BFT, with a 43% chance of incorrect application and unsuccessful euthanasia. The NPCB device was 100% effective across all age groups (i.e., >150 g) in all trials. Mechanical cervical dislocation failure occurred for one pre-weaned kit and two adults (Table 2). These animals were outside the weight range with which the operator had experience and the method required either plate adjustment (for the preweaned kit) or increased applied force (for the two larger does) to be effective. The minimum size and age evaluated for MCD was 150 g or two weeks of age. Although it did not occur in this study, the MCD operator cautioned that too much force can be used when first learning the method, resulting in decapitation (deemed a success in terms of euthanasia but esthetically displeasing to the operator). There was no association between age and probability of method failure when evaluated across all methods, χ^2^ (2, *n* = 170) = 2.66, *p* = 0.27. When analysed within each method, there was also no association between age and probability of method failure (BFT, χ^2^ (2, *n* = 58) = 4.66, *p* = 0.1; MCD, χ^2^ (2, *n* = 49) = 5.70, *p* = 0.06).

For rabbits successfully euthanized on a first attempt, all reflexes were absent when first checked immediately after method application. Although the pupillary reflex was used throughout the study, assessments were conducted outdoors (i.e., a brightly lit environment), limiting its utility. Rhythmic breathing was a reliable indicator of return to sensibility, predicting the return of other reflexes as well as purposeful movement. Vocalizations occurred when a method was incorrectly applied and the rabbit was still sensible, or occasionally, when an operator restrained a rabbit by its back legs prior to application of BFT.

The average total convulsion time for BFT, the NPCB device, and MCD was 50 s ± 7, 62 s ± 4, and 58 s ± 7, respectively. There was a difference between groups as determined by one-way ANOVA [F(2,140) = 5.03, *p* = 0.008]. A post hoc Tukey’s test indicated that total time convulsing following BFT was shorter than for the NPCB device (*p* = 0.01) and MCD (*p* = 0.03). Overall, clonic and tonic convulsions in rabbits did not follow a clear pattern or transition and occurred variably. Convulsion occurrence and strength seemed to depend on the health status of the animal, with healthier animals having more aggressive convulsions, although this was not specifically scored in this study. A pattern predictive of a return to sensibility was convulsions suddenly stopping versus slowly fading out. No convulsions occurred when rabbits were sensible. Cardiac arrest occurred at 160 s ± 5 after the method was applied with a range across all successful euthanasia of 70 s to 388 s. Heartbeat was challenging to palpate and auscultate for young pre-weaned rabbits, emaciated rabbits, and for rabbits euthanized by BFT.

After application of the NPCB device, subcutaneous swelling was immediately evident at the site of application. No distal vertebral fractures or hip luxations were noted after application of MCD, and dislocations were only detectable with palpation. Rabbits were difficult to restrain for BFT and this difficulty in safely restraining them may have attributed to method failure. Rabbits were easily restrained for application of the NPCB device and the relatively large bolt and barrel of the device to the size of the head of the rabbit allowed for easy alignment.

### 3.2. Survey Radiograph Findings

Survey radiographs of the four growers euthanized by the NPCB device demonstrated a depressed cranium, some with skull fragments embedded in the brain and all rabbits had noticeable swelling from subcutaneous hemorrhage (Figure 3). Those euthanized by BFT varied in the degree of damage to the cranium. Two rabbits demonstrated limited, focal fractures, one rabbit had depressed fragments (i.e., compression fracture), and one rabbit (Figure 4) had two significant cranium fractures. The radiographs of the four growers euthanized by MCD showed consistent trauma resulting from complete dislocation between the base of the skull and the first vertebrae with no vertebral fractures or other dislocations (Figure 5).

### 3.3. Macroscopic Assessment of Tissue Damage

Macroscopic evaluations consistently demonstrated marked brain hemorrhage for rabbits euthanized with the NPCB device. Swelling of the tissue above the skull (subcutaneous hematoma) resulted in high macroscopic subcutaneous hemorrhage scores (Figure 3). Rabbits euthanized by MCD had a small dislocation gap but a large amount of cervical hemorrhage resulting from ruptured blood vessels (Figure 5). Some rabbits euthanized by MCD had minor skull fractures of the occipital bone. This is the location the device applies pressure during application while holding the rabbit’s head in place, and in such cases a clean dislocation still occurred. Macroscopic damage resulting from BFT varied and was often not obvious (Figure 4). Damage, including fractures, was commonly noted in areas other than the targeted location for BFT, such as the back, shoulder blades, and nasal cavity. There was a statistically significant increase in brain hemorrhage score for the NPCB device compared to the other two methods, χ^2^ (2, *n* = 154) = 120.82, *p* < 0.001 for macroscopic subcutaneous and χ^2^ (2, *n* = 154) = 82.36, *p* < 0.001 for subdural hemorrhages, respectively. There was a statistically significant difference between all three methods for skull fractures, χ^2^ (2, *n* = 154) = 93.10, *p* < 0.001, with the NPCB device scoring the highest, followed by BFT then MCD.

Comparing between age groups, skull fracture scores were higher for pre-weaned kits compared to adult rabbits across all methods, χ^2^ (2, *n* = 154) = 9.60, *p* = 0.008, whereas no differences were noted between the other age groups. Subcutaneous and subdural macroscopic hemorrhages were not significantly different between age groups across all methods. Comparisons for age within treatment groups indicated that within the BFT group, pre-weaned kits scored significantly higher than adults for macroscopic subcutaneous hemorrhage, χ^2^ (2, *n* = 45) = 8.19, *p* = 0.02, and for skull fractures, χ^2^ (2, *n* = 45) = 9.81, *p* = 0.006. Within the NPCB group, adults had a significantly lower skull fracture score than pre-weaned kits and growers, χ^2^ (2, *n* = 63) = 9.81, *p* = 0.001. Within the MCD group there was less macroscopic subdural hemorrhage in the growers than the adults, χ^2^ (2, *n* = 46) = 8.32, *p* = 0.02. These were the only statistically significant differences among age groups.

### 3.4. Microscopic Brain Hemorrhage Scores

Histologic analysis of the three sections from each brain indicated that rabbits euthanized with the NPCB device had the highest average subdural and parenchymal hemorrhage scores across brain sections, except for the parenchymal score in the hindbrain (Table 3). Comparing brain sections from the hindbrain, BFT scored significantly lower than other methods for subdural hemorrhage hindbrain scores, χ^2^ (2, *n* = 38) =9.06, *p* = 0.01. There were no significant differences in hindbrain parenchymal hemorrhage scores between methods. Comparing brain sections from the mid-brain, the NPCB device scored significantly higher than other methods for mid-brain subdural hemorrhage, χ^2^ (2, *n* = 38) = 15.03, *p* = 0.001, and scored significantly higher than BFT for mid-brain parenchymal hemorrhage, χ^2^ (2, *n* = 38) = 9.22, *p* = 0.010. Comparing brain sections from the cortex, there was no significant difference in cortex subdural hemorrhage scores between methods, but NPCB cortex parenchymal hemorrhage scores were significantly higher than other methods, χ^2^ (2, *n* = 38) = 19.957, *p* < 0.001.

Within the MCD group, comparing all brain section scores, there was significantly more microscopic parenchymal hemorrhage in the hindbrain than in the cortex, χ^2^ (2, *n* = 39) = 12.39, *p* = 0.002 (Table 3). Within the NPCB group, comparing all brain section scores, there was significantly more parenchymal hemorrhage in the cortex than in the hindbrain, χ^2^ (2, *n* = 39) = 9.584, *p* = 0.008. There were no other significant differences within groups for microscopic brain hemorrhage scores. 

The highest overall microscopic subdural and parenchymal hemorrhage score for each brain, regardless of brain section origin, indicated that all brains had some form of hemorrhage. Average highest overall subdural and parenchymal hemorrhage scores were significantly higher for the NPCB device than BFT, χ^2^ (2, *n* = 38) = 12.31, *p* = 0.001, χ^2^ (2, *n* = 38) = 10.94, *p* = 0.005 respectively (Table 3). There were no significant differences between other methods or between highest overall scores and age groups across all methods. The most common highest microscopic subdural hemorrhage score across methods was 2 for BFT, 4 for the NPCB device, and 3 for MCD. The most common highest parenchymal hemorrhage score was 0 for BFT, 3 for the NPCB device, and 2 for MCD. Comparisons made within treatment groups indicated that the only significantly different overall microscopic hemorrhage scores between age groups were within the BFT group. Growers euthanized by BFT had significantly higher subdural hemorrhage scores than adults euthanized by the same method, χ^2^ (2, *n* = 12) = 6.72, *p* = 0.04.

## 4. Discussion

Our results demonstrated that the NPCB device was 100% effective for euthanizing all age groups of commercial meat rabbits tested (i.e., >150 g) and that the MCD device was also a highly effective technique with an overall 6% failure rate, largely attributable to inexperience with one specific weight range. Blunt force trauma had the highest failure rate with 13 of 63 animals (22%) unsuccessfully euthanized, including a 43% failure rate (8 of 14) in adult rabbits. Based on these findings we cannot recommend blunt force trauma as a euthanasia method for commercial meat rabbits. 

Several reasons may have accounted for BFT failures, one of which is insufficient force. Li et al. [12] found that a difference of 150 kPa of force applied during machine operated BFT altered the chance of mortality in 2 kg rabbits. Rabbits in their mild injury group were also less likely to die from their brain injuries with only 10% mortality compared to 60% mortality in the marked injury group. Although this may have been a factor in our study, based on our observations, a more likely cause for method failure is a lack of accuracy in applying the method to the correct anatomic location. Erasmus et al. [7] found that it is more challenging to apply BFT accurately to broiler turkeys than adult turkey toms and attributed this to a smaller target area. However, our research found more microscopic brain hemorrhage following BFT application to growers than adult rabbits. Difficulty in safely restraining the rabbits was observed to be the cause limiting accuracy when applying BFT. The NPCB device has been termed a controlled form of BFT because of the similar trauma goal. Use of a device such as the NPCB greatly improved accuracy. Reasons for high accuracy for the NPCB device include minimal restraint needed and easy alignment and application.

The device used for MCD also resulted in a high degree of accuracy with clean dislocation at the base of the skull. Dislocations were only slightly palpable, limiting the ability to check for correct application other than through assessment of sensibility, although subcutaneous cervical swelling and hemorrhage was often marked after application. Gap distances in poultry after application of manual cervical dislocation are significantly larger (~5 cm) than those found in this study making them easier to palpate [15]. A small gap distance for rabbits could be related to their large neck muscle mass. However, Martin et al. [15] found that heavier birds had a larger dislocation than lighter birds, and this was not correlated with age. Gap distance is more likely related to anatomic differences between species. This device allowed cervical dislocation to be used on rabbits weighing >1 kg, for which manual cervical dislocation is not recommended because of technical difficulty [6,9]. 

The immediate insensibility noted following MCD in rabbits in this study is very different from findings in poultry, in which brainstem reflexes are reported to persist for several seconds after application [7,15]. Erasmus et al. [16] found that turkey hens were not immediately insensible following CD, possibly because of cervical crushing, rather than dislocation, and corresponding to lower MH scores. The mean microscopic hemorrhage scores for rabbits euthanized successfully by MCD in this study were 2.1 (±0.2) for subdural and 0.9 (±0.2) for parenchymal, compared to 0.5 (±0.5) for subarachnoid and 0 (±0) for parenchymal for turkey hens after cervical crushing [16]. Immediate insensibility following CD may be a species-specific response or a response specific to the device used. This is the first study to assess a wall-mounted device for MCD in any species. It has been suggested that a concussive force is needed during CD to cause immediate insensibility [17]. Within the MCD group of rabbits, microscopic scores of parenchymal hemorrhages were higher in the hindbrain than the cortex, as expected based on the area where the device causes damage. This suggests that sufficient force is applied to cause immediate and significant vessel disruption and hemorrhage at a critical area responsible for maintaining sensibility.

There was a significant difference between all three euthanasia methods in the number of skull fractures. No skull fractures were predicted to occur during MCD in accordance with proper technique. Although no vertebral fractures were seen, some rabbits had minor occipital bone fractures caused by pressure of the device during downward application force. The mild skull fractures noted rarely following application of MCD are unlikely to contribute to poor animal welfare since insensibility was immediate after application, which occurs <2 s after sliding the head into the V-shaped opening. The difference in skull damage between BFT and NPCB rabbits is in line with minimal microscopic brain damage with BFT, even when euthanasia was successful. Across all euthanasia methods, pre-weaned kits had more skull fractures than adults. Skull thickness correlates with age and this finding likely occurred because of the thinner skulls of younger rabbits, which fractured more readily. This supports our recommendation for altering pressure settings for the NPCB device according to animal age and degree of skull thickness. Finnie et al. [18] used a different model of the NPCB device and compared the effects on pig and lambs, finding that less skull damage occurred in pigs than lambs, also related to skull thickness.

The NPCB device caused significantly more focal macroscopic subcutaneous hemorrhage than the other two methods. The strength of subcutaneous hemorrhage findings for evaluating euthanasia efficacy is minimal compared to the other measurements, based on research evaluating cervical crushing in turkeys [17]. Sensible turkeys with no direct damage to the brain still had a significant amount of macroscopic subcutaneous hemorrhage. The predicted reason was ruptured carotid arteries in the neck causing blood to flow under the scalp. This might be the case for both MCD and BFT in this study and not reflective of direct damage to the brain. Unlike macroscopic subcutaneous hemorrhage, scores for macroscopic subdural hemorrhage, and microscopic subdural and parenchymal hemorrhage have been used as indicators of traumatic brain injury that are likely to be associated with insensibility and eventual mortality [8,17,18]. Specifically, in rabbits, a higher degree of microscopic parenchymal hemorrhage was associated with mortality after BFT [12]. Rabbits euthanized with the NPCB device had more parenchymal cortical hemorrhage than in the hindbrain. Piglets euthanized with the NPCB device similarly had significantly more parenchymal cortical hemorrhage than in other brain sections [8,11], which is the area of the brain the device directly targets when applied properly in pigs and rabbits. Overall rabbits euthanized with the NPCB device had a mean microscopic parenchymal hemorrhage score of 2.1 ± 0.2, which is comparable to the mean parenchymal hemorrhage score for other species euthanized with the NPCB device [8,11,16]. The NPCB device caused significantly greater macroscopic subdural hemorrhage and microscopic subdural and parenchymal hemorrhage scores than the other methods. Within the BFT group, scores for microscopic parenchymal and subdural hemorrhage were significantly greater for the growers than the adults, again, likely related to skull thickness and increased failure rates in adults. Regardless of the degree of trauma noted, clonic and tonic convulsions in rabbits did not follow any clear pattern or transition as reported for other species, such as piglets, where clonic convulsions begin first, and are followed by tonic convulsion [8,11].

Esthetics are an important consideration when evaluating euthanasia methods as euthanasia can have detrimental mental effects on the operator and observers [19]. Increasing the esthetics of a method also increases the operator’s comfort and confidence with euthanasia, and may result in euthanasia being applied in a more timely fashion on-farm [4]. The NPCB device requires minimal animal restraint, is associated with minimal external bleeding, and distances the operator from the task, eliminating the need for direct physical force to induce cerebral trauma. Operators using a NPCB device for piglets rated the device highly (8.7/10), indicating that it was easy to use and more esthetically pleasing than BFT [8]. In rabbits, manual BFT is associated with a high failure rate, difficult restraint, significant physical force is required, and it results in marked external bleeding, making it unpleasant esthetically and resulting in a potential biosecurity risk. Mechanical cervical dislocation still requires some physical force, and when first learning the technique, may result in decapitation or method failure. We recommend that producers first practice on cadavers of different sizes with this technique to gain competency. In experienced hands, the MCD device can be applied rapidly and the technique results in immediate insensibility of rabbits >150 g.

Among the three euthanasia methods evaluated in this study, there were differences in the training and experience operators had with their preferred methods. None of the producers had ever been formally trained in BFT, but all had been using the technique for eight or more years. The operator for the MCD device had been using this method for six months prior to this study and was self-taught based on an online video. The operator of the NPCB device had received training from an experienced NPCB device operator and had practiced the technique on rabbit cadavers during the pilot study. With any physical method of euthanasia there can be a learning curve [6] and this is thought to be the reason for the three MCD failures. These animals were outside the weight range that the operator had experience with and two required increased force for complete cervical dislocation. Our study suggests that the NPCB device requires minimal training for proficiency. Limitations of this study include that operators were not randomly assigned to methods, that methods of BFT were variable between farms, and all techniques could not be conducted on all farms. Instead operators conducted the methods that they were experienced with on their own animals. This is a drawback of using an on-farm study design, but does highlight the relevance of the results found for BFT within the meat rabbit industry, which were troubling regardless of technique used, operator or farm. The study design also meant that scoring for insensibility and macroscopic damage could not be blinded, although radiographic and microscopic assessments were blinded and were highly correlated to the clinical and gross findings.

Cost and operator safety are important considerations to ensure that validated euthanasia methods are practical. Blunt force trauma does not require a specific tool, but safety is a significant concern, as difficulty in restraining a rabbit could result in self injury. External bleeding and tissue discharge are also biosecurity safety concerns for BFT. The MCD device can be made on-farm for ~$35 USD or purchased for ~$90 USD (Rabbit Wringer, http://www.rabbitwringer.com, West Grove, PA, USA). This device accommodates different sizes of rabbits with an adjustable neck plate. The device does not pose a physical safety or biosecurity risk for the operator. The NPCB device used was the Zephyr-E, an expensive device at ~$900 USD (Bock Industries, Philipsburg, PA, USA). Restraint of pre-weaned kits increases the proximity of the device to the hand leading to a potential safety risk. The manufacturer of the device indicates that accidental misfire of the device to the hand would result in bruising. Our results are specific to this particular instrument design (including the modified end of the bolt) and the size of the rabbit. Re-validation would be required if modifications to the device were attempted or if the device was applied to rabbits of a breed other than the New Zealand white-like rabbits and outside the weight range of 0.15 to 3.5 kg evaluated in this study.

## 5. Conclusions

Our findings indicate that blunt force trauma is neither a humane nor an esthetically acceptable method for killing meat rabbits of any age and it is not recommended for on-farm euthanasia of rabbits. Both the non-penetrating captive bolt device and the mechanical cervical dislocation device used in this study provided effective and humane single step methods for euthanasia of rabbits >150 g.

## Figures and Tables

**Figure 1 animals-07-00100-f001:**
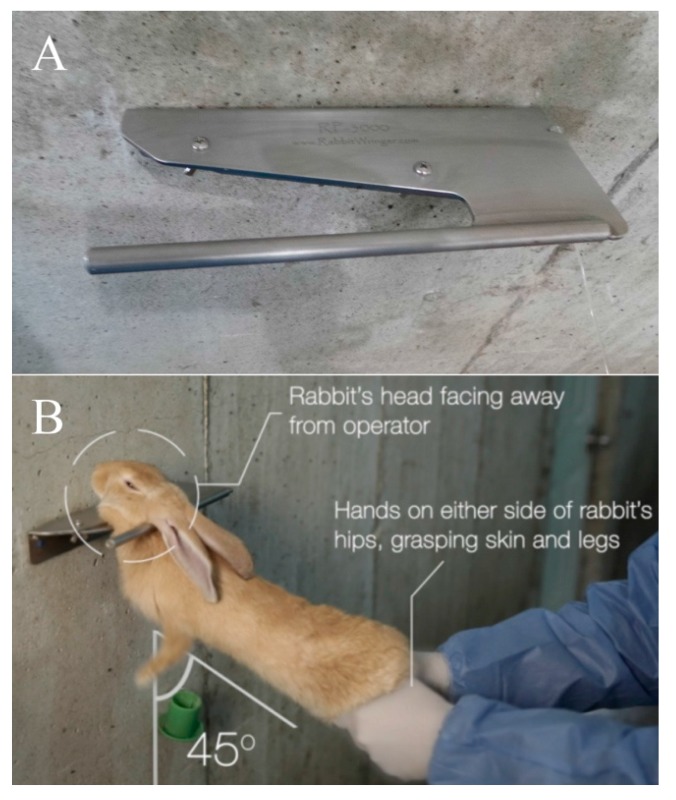
(**A**) Device employed for mechanical cervical dislocation; and (**B**) positioning of animal for correct application.

**Figure 2 animals-07-00100-f002:**
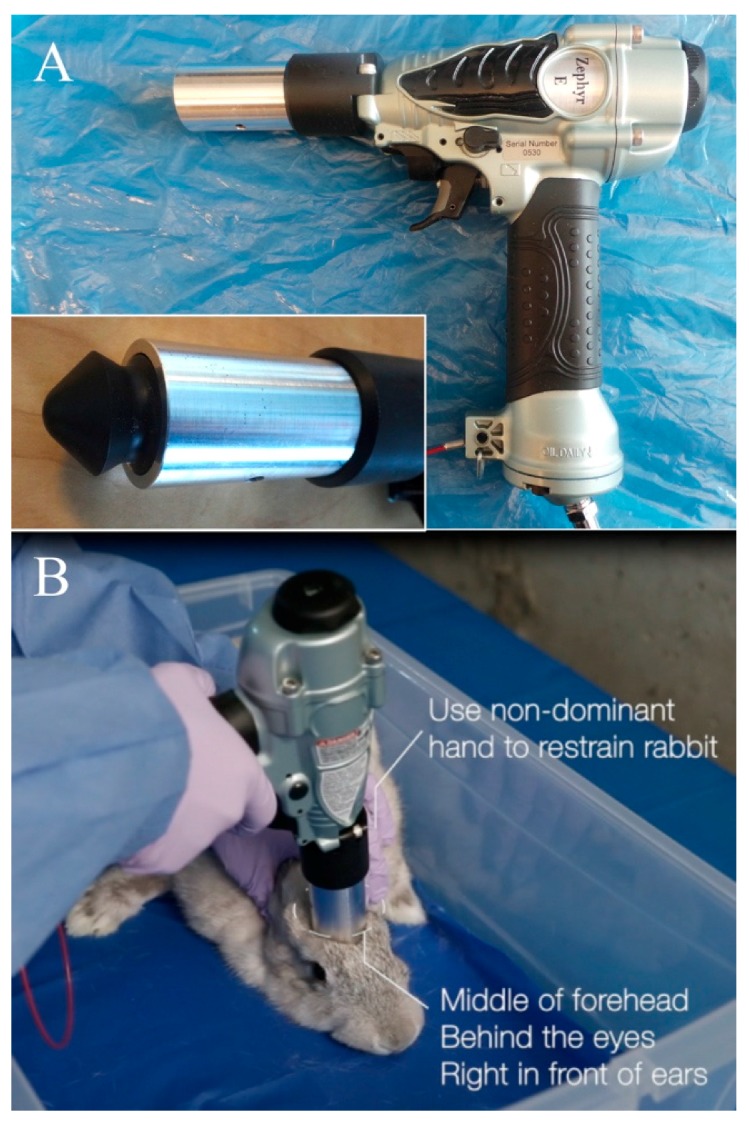
The non-penetrating captive bolt device used in study trials. (**A**) The device with the safety pin removed. Inset: The bolt at its full extension from the barrel; (**B**) Correct location and handling for applying the non-penetrating captive bolt device to a rabbit’s head.

**Figure 3 animals-07-00100-f003:**
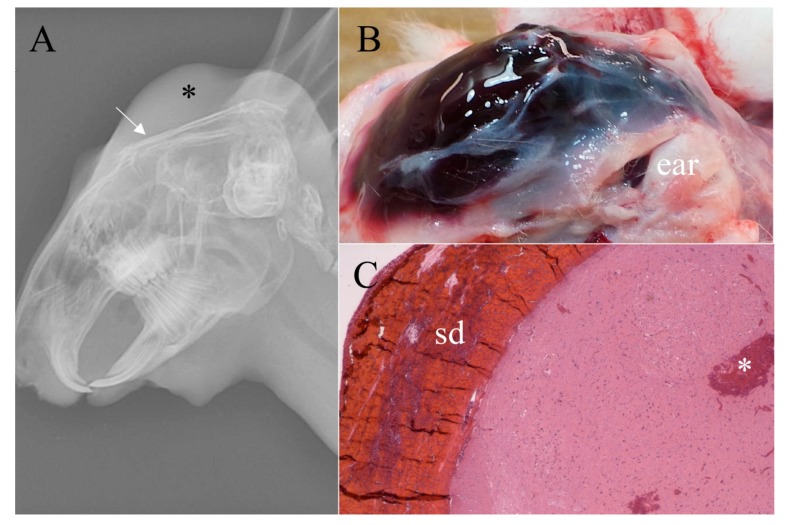
Eight week-old grower rabbit euthanized with the non-penetrating captive bolt device. (**A**) Survey radiograph demonstrating fractured and depressed cranium (arrow) and marked subcutaneous swelling and hemorrhage (*); (**B**) Skin reflected to demonstrate marked subcutaneous hemorrhage completely covering the area from the eyes to the base of the skull (gross score of 4/4); (**C**) Photomicrograph of midbrain demonstrating marked subdural hemorrhage (sd) (histologic score of 4/4) and moderate parenchymal hemorrhage (*) (histologic score of 3/4) (H & E, ×20).

**Figure 4 animals-07-00100-f004:**
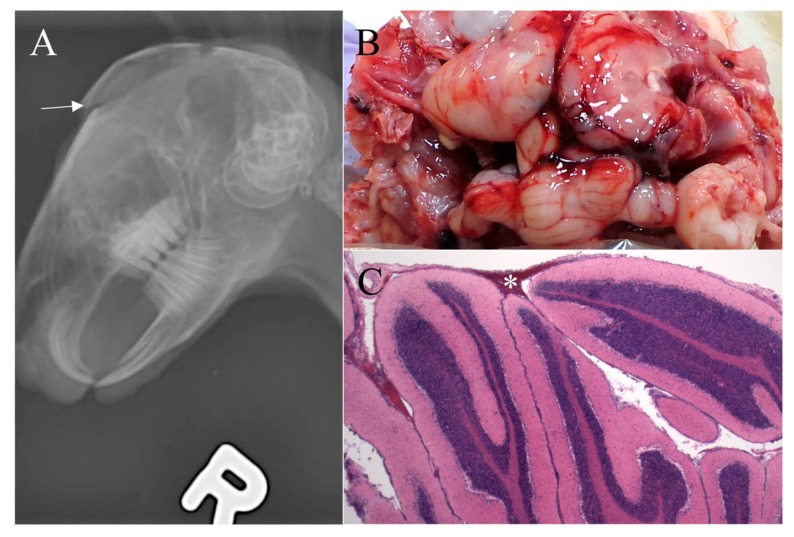
Six week-old grower rabbit euthanized by blunt force trauma. (**A**) Survey radiograph demonstrating fractured cranium (arrow); (**B**) Minimal subdural dorsal hemorrhage covers less than 25% of the brain surface (gross score of 1/4); (**C**) Photomicrograph of cerebellum (demonstrating mild subdural hemorrhage (*) (histologic score of 2/4) and no parenchymal hemorrhage (histologic score of 0/4). No subdural or parenchymal hemorrhage were noted in the midbrain. (H & E, ×20).

**Figure 5 animals-07-00100-f005:**
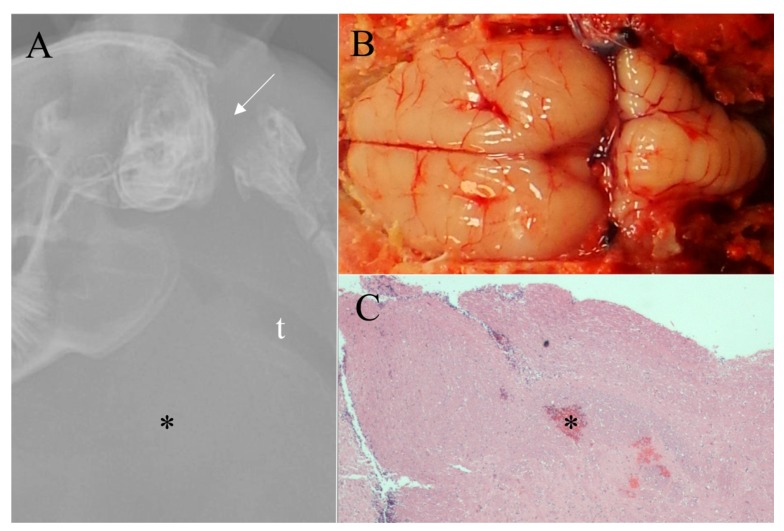
Eight week-old grower rabbit euthanized by mechanical cervical dislocation. (**A**) Survey radiograph shows a clean dislocation between the base of the skull and the first vertebrae (arrow) and marked cervical hemorrhage (*), t = trachea; (**B**) Grossly, no subdural dorsal hemorrhage was noted (gross score of 0/4); (**C**) Photomicrograph of midbrain demonstrating a lack of subdural hemorrhage (histologic score of 0/4) and minimal parenchymal hemorrhage (*) (histologic score of 1/4) (H & E, ×20).

**Table 1 animals-07-00100-t001:** Scoring system for grading skull fractures and macroscopic subcutaneous and subdural hemorrhage in rabbits euthanized by different methods. ^1^

Score	Fracture Score Description	Hemorrhage Score Description
0	No fracture, intact skull	No hemorrhage
1	Hairline fracture, no separation of bone	<25% of surface area covered
2	1 to 2 complete fully separated fractures or single depressed fracture	26 to 50% of surface are covered
3	More than just a single depressed fracture, 3 to 5 complete fractures	51 to 75% coverage
4	>5 complete fractures	76% to complete coverage

^1^ Adapted from Casey-Trott et al. [13] and Veltri and Klem [14].

**Table 2 animals-07-00100-t002:** Number of cull rabbits per age group euthanized by different methods included in each stage of the trial.

Method	Age Group ^1^	Total Euthanized	Body Weight, kg (mean ± SE)	No. Successfully Euthanized (%)	No. Radio-Graphed	No. Gross Scored	No. Microscopic Scored
Blunt force trauma (BFT)	Pre-weaned kits	23	0.1 ± 0.03	20 (87)	1	20	3
	Growers	21	1.6 ± 0.1	17 (81)	3	17	3
	Adults	14	3.6 ± 0.3	8 (57)	0	8	6
	**Overall**	**58**	**1.5 ± 0.2**	**45 (78)**	**4**	**45**	**12**
Non-penetrating captive bolt device (NPCB)	Pre-weaned kits	17	0.2 ± 0.04	17 (100)	0	17	3
	Growers	26	1.4 ± 0.1	26 (100)	4	26	4
	Adults	20	3.3 ± 0.2	20 (100)	0	20	6
	**Overall**	**63**	**1.7 ± 0.2**	**63 (100)**	**4**	**63**	**13**
Mechanical cervical dislocation (MCD)	Pre-weaned kits	9	0.4 ± 0.1	7 (78)	0	7	3
	Growers	25	1.3 ± 0.1	25 (100)	4	25	4
	Adults	15	3.3 ± 0.2	14 (93)	0	14	6
	**Overall**	**49**	**1.7 ± 0.2**	**46 (94)**	**4**	**46**	**13**
Grand total		170	1.6 ± 0.1	154 (91)	12	154	38

^1^ Pre-weaned kits = 0 to 5 weeks, Growers = 6 to 12 weeks, Adults ≥ 12 weeks.

**Table 3 animals-07-00100-t003:** Mean (±SE) subdural (SD) and parenchymal (P) microscopic hemorrhage scores across euthanasia methods and brain locations.

Method	Brain Section ^1^						Mean Scores	
	Hindbrain		Mid-Brain		Cortex			
	SD	P	SD	P	SD	P	SD	P
Blunt force trauma (BFT)	1.4 ^a^ ± 0.4	0.8 ^a^ ± 0.3	1.6 ^b^ ± 0.4	0.6 ^b^ ± 0.3	1.7 ^a^ ± 0.4	0.6 ^b^ ± 0.4	1.6 ^b^ ± 0.2	0.7 ^b^ ± 0.2
Non-penetrating captive bolt (NPCB)	3.0 ^b^ ± 0.3	1.5 ^a,bb^ ± 0.3	3.4 ^a^ ± 0.2	1.8 ^a^ ± 0.3	3.0 ^a^ ± 0.3	2.9 ^a,aa^ ± 0.3	3.1 ^a^ ± 0.1	2.1 ^a^ ± 0.2
Mechanical cervical dislocation (MCD)	2.8 ^b^ ± 0.3	1.6 ^a,aa^ ± 0.2	1.8 ^b^ ± 0.3	0.9 ± 0.3	1.7 ^a^ ± 0.4	0.3 ^b,bb^ ± 0.2	2.1 ± 0.2	0.9 ± 0.2

Different single letter subscripts denote a significant difference between methods for column comparisons of average score, a ≠ b, *p* < 0.05. Different double letter subscripts denote a significant difference for a type of brain hemorrhage between brain sections for row comparisons of average score, aa ≠ bb, *p* < 0.05. See methods for a complete description of scoring. ^1^ Each brain was divided into three coronal sections, *n* = 38 brains (blunt force trauma = 12, non-penetrating captive bolt = 13, mechanical cervical dislocation = 13). 114 brain sections (hindbrain = 38, mid-brain = 38, cortex = 38).

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
