# Peer review of "Efficacy of Blunt Force Trauma, a Novel Mechanical Cervical Dislocation Device, and a Non-Penetrating Captive Bolt Device for On-Farm Euthanasia of Pre-Weaned Kits, Growers, and Adult Commercial Meat Rabbits"

_animals, 2017, doi:10.3390/ani7120100_

Round 1

Reviewer 1 Report

Overall this a very interesting paper on a topic which requires attention to improve welfare on a wider scale. A few specific changes required, but two general comments are: the definition of assisted manual cervical dislocation is an oxymoron. Manual is defined as cervical dislocation performed by a human operator without the use of tools or aids and is based solely on the operator’s skill and technique. Mechanical cervical dislocation involves the use of tools and aid and is defined as such by AVMA, EU directives and HSA. The use of a rabbit wringer is mechanical cervical dislocation. The authors regularly confuse reflexes and behaviours which are used to define brain death, and not insensibility alone. This needs to be addressed as it affects how the authors interpret the results.

L25: It may be clearer to say “The non-penetrating captive was the most effective with a 100% 25 success rate and could be used on all rabbits weighing more than 150g.

L31: please state that the number of rabbits was unbalanced across kill treatments

L60: please reference examples of mechanical cervical dislocation use in poultry and mice

L66: please reference efficacy of NPCB device in poultry

L80: unbalanced design, what was the spread of the 170 euthanised rabbits across the three farms and across the 28 days, as well as rabbit age – risk of confounds

L81: states sample size calculations done – please report these and what estimates/assumptions these are based on

L88: operator, method and rabbit age are confounded

L91: method and farm are confounded

L88: How many operators were part of the study and how was this spread across various factors e.g. method, farm etc.

L103: How rapid is the two shots? Does the first shot cause immediate insensibility – if not, this is a welfare issue

L114: also potential welfare risk if non-fatal application occurred - pain associated with damage to eyes and sinuses.

L127: Has the zephyr been reliability tested in terms of consistency pressure across multiple shots? Please provide details.

Figure 2: Does the device need to be pushed onto the animal’s head prior to firing? If so, how much force is required, potential welfare issue. Needs describing in methods and discussion later on if this is the case.

L136 Pupillary reflex and corneal reflex are indicators of brain death (and therefore complete insensibility) not insensibility alone.

L138 “reliable response” is only in a conscious unharmed animal, following traumatic brain injury, movement and reactions could be in terms of motor function impaired, but actual reliable and definite signs of insensibility are not possible during killing methods from behaviour alone.

L149 absence/presence of rhythmic breathing is an indicator of brain death and not insensibility alone.

L150 “purposeful” vocalisations – how was this determined by the authors? How was the motivation or cause of the behaviour identified? Please remove the word “purposeful”

L189 This paragraph describing the numbers of rabbits for macroscopic analysis is not that clear – please rephrase for clarity

L204 – 219 Statistical analysis is too limited and does not report on how any confounds were addressed. Statistical tests are a little too simple – GLM or GLMMs should have been used to incorporate random and fixed effects together. As a result, all reported results are limited in terms of what you can actually conclude, e.g. operator, method, farm etc. are confounded with one another.

L306 with only a small dislocation gap, the likelihood of rupturing major blood vessels e.g. carotid arteries, is unlikely. The rupturing of these is required for cessation of blood flow directly to the brain to cause rapid brain death and loss of consciousness.

Author Response

Comments and Suggestions for Authors

Overall this a very interesting paper on a topic which requires attention to improve welfare on a wider scale. A few specific changes required, but two general comments are: the definition of assisted manual cervical dislocation is an oxymoron. Manual is defined as cervical dislocation performed by a human operator without the use of tools or aids and is based solely on the operator’s skill and technique. Mechanical cervical dislocation involves the use of tools and aid and is defined as such by AVMA, EU directives and HSA. The use of a rabbit wringer is mechanical cervical dislocation. 

We were attempting to define a novel but passive device, whereas most mechanical CD devices are applied in a more active fashion. We agree that this may be confusing for the reader and have changed throughout.

The authors regularly confuse reflexes and behaviours which are used to define brain death, and not insensibility alone. This needs to be addressed as it affects how the authors interpret the results.

Clarified in text, as requested. Please note that the authors are not confused about the definitions of each. We emphasize that multiple parameters are needed for evaluation and both reflexes and behaviours were assessed in this study. We have clarified the use of behavioural vs reflex assessment in the methods.

L25: It may be clearer to say “The non-penetrating captive was the most effective with a 100% 25 success rate and could be used on all rabbits weighing more than 150g.

Changed, as suggested.

L31: please state that the number of rabbits was unbalanced across kill treatments

There was insufficient room to elaborate on this in the 200 word abstract; however, we have added this information to the methods and discussion sections, as suggested.

L60: please reference examples of mechanical cervical

Added, as suggested.

L66: please reference efficacy of NPCB device in poultry

Added, as suggested.

L80: unbalanced design, what was the spread of the 170 euthanised rabbits across the three farms and across the 28 days, as well as rabbit age – risk of confounds

Please see Table 2. Efficacy of technique outcomes was compared between farms by age group (only for BFT and NPCD) before combining information.

L81: states sample size calculations done – please report these and what estimates/assumptions these are based on

Added, as requested.

L88: operator, method and rabbit age are confounded

This was a field study and thus convenience sampling had to be used. Researchers visited each farm 1-2x/week, depending on producer availability) over the course of the study and could only use the animals available at each visit. This has been clarified in the methods.

L91: method and farm are confounded

The purpose of this study was to compare the NPCB device to two other techniques commonly in use on farm. We only wanted to assess techniques that producers were comfortable using, to ensure that any failures noted were not a result of technique novelty. This approach has been used in other published euthanasia studies (e.g., Casey-Trott et al, 2014).

L88: How many operators were part of the study and how was this spread across various factors e.g. method, farm etc.

Added details to methods. Three producers (1 per farm) and the researcher using the NPCB device.

L103: How rapid is the two shots? Does the first shot cause immediate insensibility – if not, this is a welfare issue

Clarified, as requested. Discharges are immediate. It is likely that that the second discharge is not needed in rabbits but we applied according to published recommendations for pigs and poultry and those from abattoir personnel.

L114: also potential welfare risk if non-fatal application occurred - pain associated with damage to eyes and sinuses.

Application of the NPCB device resulted in a fatal outcome in 100% of the cases and insensibility was immediate.

L127: Has the zephyr been reliability tested in terms of consistency pressure across multiple shots? Please provide details.

Yes, please see http://www.bock-industries.com/ for engineering specifications. This is a commercially validated device.

Figure 2: Does the device need to be pushed onto the animal’s head prior to firing? If so, how much force is required, potential welfare issue. Needs describing in methods and discussion later on if this is the case.

Clarified in Methods. The device rests gently on the animals head but must be in direct contact.

L136 Pupillary reflex and corneal reflex are indicators of brain death (and therefore complete insensibility) not insensibility alone.

We respectfully disagree. Pupillary light reflex and corneal reflex are indicators of insensibility and lack of response can be seen in heavily sedated (pupillary) and anesthetized (both) animals – not just dead animals.

L138 “reliable response” is only in a conscious unharmed animal, following traumatic brain injury, movement and reactions could be in terms of motor function impaired, but actual reliable and definite signs of insensibility are not possible during killing methods from behaviour alone.

We agree, which was why we used multiple indicators for assessment in this study. This has been further clarified in the methods.

L149 absence/presence of rhythmic breathing is an indicator of brain death and not insensibility alone.

We are unclear as to what the comment is. Rhythmic breathing can disappear after stunning and return to rhythmic breathing is widely reported in improperly stunned poultry, pigs, etc.

L150 “purposeful” vocalisations – how was this determined by the authors? How was the motivation or cause of the behaviour identified? Please remove the word “purposeful”

Corrected, as suggested.

L189 This paragraph describing the numbers of rabbits for macroscopic analysis is not that clear – please rephrase for clarity

Corrected, as suggested.

L204 – 219 Statistical analysis is too limited and does not report on how any confounds were addressed. Statistical tests are a little too simple – GLM or GLMMs should have been used to incorporate random and fixed effects together. As a result, all reported results are limited in terms of what you can actually conclude, e.g. operator, method, farm etc. are confounded with one another.

We agree that analysis of field studies can be challenging. Because of this, we consulted extensively with a statistician for this study and proceeded largely with nonparametric statistical methods based on this advice. If anything, the conclusions drawn are a conservative measure of differences between groups and potentially underestimate problems with on-farm killing, especially for BFT.

L306 with only a small dislocation gap, the likelihood of rupturing major blood vessels e.g. carotid arteries, is unlikely. The rupturing of these is required for cessation of blood flow directly to the brain to cause rapid brain death and loss of consciousness.

Thank-you for your comment. We observed a large amount of cervical hemorrhage resulting from ruptured blood vessels as mentioned. While the dislocation gap did not appear large in the radiographs, the force and technique did cause rupturing of major blood vessels.

Reviewer 2 Report

In this paper, three methods for on-farm euthanasia of rabbits of different ages are compared and evaluated scientifically. In terms of animal welfare it is crucial to provide practicable on-farm euthanasia protocols, which imply minimal pain and distress prior to and during the procedure, and which result in a rapid loss of sensibility prior to death. In contrast to e.g. poultry, such methods have not been evaluated scientifically for rabbits underlining the novelty of the present study. It is also worth mentioning that the authors only use cull rabbits for their investigation (instead of “healthy” laboratory animals), and that they confirm their observations (concerning the immediate loss of sensibility/effectiveness of the method) by means of macroscopic and microscopic pathological findings.

I think the paper is of high practical relevance since it provides scientific proof for the inadequacy of a commonly used euthanasia method for rabbits (BFT), and also offers feasible alternatives which can be applied on-farm (AMCD, NPCB).

However, there are some minor items that have to be addressed.

Throughout the manuscript, you mainly write numbers in figures (e.g. L. 410 “…3 euthanasia methods…”; L. 421 “…the other 2 methods”). When not followed by measurement units, I would prefer numbers up to ten or twelve written in words. However, as I think animals has no house style concerning this, you can decide yourself but be consistent (e.g. L. 455 “…the three euthanasia methods…”).

Line remarks:

L. 46:     Introduction: In Europe, the Council Regulation (EC) No 1099/2009 governs the protection of animals at the time of killing including also provisions for emergency killing and allowed procedures for the respective animal species. Are there similar regulations in Canada which may be cited here?

L. 51:     Please define the abbreviation “AVMA”, it may not be clear for readers from other parts of the world.

L. 61:     CD devices must also provoke cerebral ischemia by causing vessel disruption. Of cause, this is associated with a rapid loss of sensibility; however, it should be emphasized here. A clean dislocation between the skull and first cervical vertebrae without vessel disruption would only result in paraplegia.

L. 73:     The rabbits involved in this study were targeted for euthanasia by the producers. Was this selection checked by a vet/scientist? Were there any animals for which veterinary treatment would have been justifiable?

L. 84:     What was the breeding company of the rabbits?

L. 133:   See comment on L. 73.

L. 151:   Based on which criteria was it decided whether the same or an alternative method was used when the first euthanasia attempt was deemed a failure?

L. 173:   Did you check inter-observer reliability statistically, for instance by calculating Kappa-coefficients?

Table 1:                You should reformat the lines of the table since it is not immediately clear to which description each score belongs. (Maybe use “align top center” for placing the text instead of “align center”). Instead of the second footer, you should consider another column for the score evaluations (mild, moderate and marked).

L. 207:   Since the Shapiro-Wilk test is very strict, have you also tested normal distribution by means of histograms r Q-Q-plots? How did you test for homogeneity of variance (a further criterion for the ANOVA)? With the procedure according to Levene?

L. 219:   Did you control for multiple testing, for instance by means of the Bonferroni correction?

L. 229:   Change “A…” to “An…”.

L. 244:   Change “aesthetically” to “esthetically” for style consistency.

L. 247:   You state that there was no association between age and probability of method failure. Would P=0.06 not indicate a (significant) trend in AMCD?

Table 2:                You should also add the abbreviations of the euthanasia methods in the respective column.

Figure 5:              I can see the trachea (t) in the radiograph, however, the cervical edema (*) is not visible. Would it be possible to adjust saturation/contrast/brightness of the picture?

Table 3:                This table is not clearly arranged. SEs should appear next to the means they refer to. Please format this table like Table 2.

Heading: “Mean (±SD)”, did you mean SE? For in L. 207 you state that all values are reported as mean±SE (Standarderror not Standarddeviation). In addition, the abbreviation for “subdural” is missing.

Footer: Change “p<0.05” to “P<0.05” for style consistency. Change “Methods” to “methods”.

            L. 358:   Change “respectfully” to “respectively”

L. 444:   You mention the positive aspects of such a device, which are unchallenged. However, could you also think of situations in which farmers may kill a rabbit overhasty with such a device, for instance in cases in which separation/veterinary treatment would have been feasible?

L. 450:   Besides the esthetically unpleasant aspect of external bleeding, you should also mention the hygienic risks here.

L. 462:   Please provide a reference for this learning curve.

L. 463:   The verb is missing in “Our study suggests that the NPCB device 463 minimal training for proficiency.”.

L. 480:   “and”.

L. 481:   Do you have any recommendations/suggestions for killing rabbits outside the tested weight range?

L. 483:   Although you explain possible welfare implications of an esthetic euthanasia method (L. 444) I would not overrate this aspect and therefore would not mention it in the conclusion. From the point of animal suffering, esthetics is completely irrelevant.

Author Response

Comments and Suggestions for Authors

In this paper, three methods for on-farm euthanasia of rabbits of different ages are compared and evaluated scientifically. In terms of animal welfare it is crucial to provide practicable on-farm euthanasia protocols, which imply minimal pain and distress prior to and during the procedure, and which result in a rapid loss of sensibility prior to death. In contrast to e.g. poultry, such methods have not been evaluated scientifically for rabbits underlining the novelty of the present study. It is also worth mentioning that the authors only use cull rabbits for their investigation (instead of “healthy” laboratory animals), and that they confirm their observations (concerning the immediate loss of sensibility/effectiveness of the method) by means of macroscopic and microscopic pathological findings.

I think the paper is of high practical relevance since it provides scientific proof for the inadequacy of a commonly used euthanasia method for rabbits (BFT), and also offers feasible alternatives which can be applied on-farm (AMCD, NPCB).

However, there are some minor items that have to be addressed.

Throughout the manuscript, you mainly write numbers in figures (e.g. L. 410 “…3 euthanasia methods…”; L. 421 “…the other 2 methods”). When not followed by measurement units, I would prefer numbers up to ten or twelve written in words. However, as I think animals has no house style concerning this, you can decide yourself but be consistent (e.g. L. 455 “…the three euthanasia methods…”).

Corrected, as suggested.

Line remarks:

L. 46:     Introduction: In Europe, the Council Regulation (EC) No 1099/2009 governs the protection of animals at the time of killing including also provisions for emergency killing and allowed procedures for the respective animal species. Are there similar regulations in Canada which may be cited here?

Added two references to address this.

L. 51:     Please define the abbreviation “AVMA”, it may not be clear for readers from other parts of the world.

Corrected, as suggested.

L. 61:     CD devices must also provoke cerebral ischemia by causing vessel disruption. Of cause, this is associated with a rapid loss of sensibility; however, it should be emphasized here. A clean dislocation between the skull and first cervical vertebrae without vessel disruption would only result in paraplegia.

Corrected, as suggested.

L. 73:     The rabbits involved in this study were targeted for euthanasia by the producers. Was this selection checked by a vet/scientist? Were there any animals for which veterinary treatment would have been justifiable?

No, animals were identified for euthanasia by the producers and this has been added to the methods for clarification. Based on our own examinations of the animals many should have been identified for culling much earlier than they were(but that is another problem altogether!). There was never an instance of an inappropriately identified cull rabbit.

L. 84:     What was the breeding company of the rabbits?

No specific company is used for breeding stock in Canada and genetics are not tracked as on some farms in the EU. Most rabbit herds are closed.

L. 133:   See comment on L. 73.

The rabbit cadavers were donated without specifying culling reasons. While we saw many different pathologies in these animals it was not necessary to track or record these for the purpose of the pilot study.

L. 151:   Based on which criteria was it decided whether the same or an alternative method was used when the first euthanasia attempt was deemed a failure?

Based on lack of insensibility after first application or signs of impeding return to sensibility.

L. 173:   Did you check inter-observer reliability statistically, for instance by calculating Kappa-coefficients?

Agreement between scorers for gross tissue hemorrhage was determined by scoring simultaneously and recording a consensus score for each animal. This has been clarified in the text.

Table 1:  You should reformat the lines of the table since it is not immediately clear to which description each score belongs. (Maybe use “align top center” for placing the text instead of “align center”). Instead of the second footer, you should consider another column for the score evaluations (mild, moderate and marked).

Corrected, as suggested.

L. 207:   Since the Shapiro-Wilk test is very strict, have you also tested normal distribution by means of histograms r Q-Q-plots? How did you test for homogeneity of variance (a further criterion for the ANOVA)? With the procedure according to Levene?

Q-Q plots were generated as part of SPSS normality and were used as a graphical interpretation of normality. Due to subjectivity in evaluating Q-Q plots, the Shapiro-Wilk test method was selected to report normality. Homogeneity of variance was also calculated as part of SPSS for parametric statistics and assessed graphically (clarified in the methods), but it is not an assumption of a nonparametric test such as the Kruskal Wallis which was used to analyze a number of parameters in this study.

L. 219:   Did you control for multiple testing, for instance by means of the Bonferroni correction?

A Bonferroni correction was applied for multiple comparisons and this has been added to the text.

L. 229:   Change “A…” to “An…”.

Corrected, as suggested.

L. 244:   Change “aesthetically” to “esthetically” for style consistency.

Corrected, as suggested.

L. 247:   You state that there was no association between age and probability of method failure. Would P=0.06 not indicate a (significant) trend in AMCD?

We used P <0.05 for significance reporting throughout, and because of the complexity of the study design chose not to report trends, which may have variable interpretations.

Table 2: You should also add the abbreviations of the euthanasia methods in the respective column.

Corrected, as suggested.

Figure 5: I can see the trachea (t) in the radiograph, however, the cervical edema (*) is not visible. Would it be possible to adjust saturation/contrast/brightness of the picture?

When cropping images to ensure similar size for reader comparison it was not possible to show the edge of the skin in this image. The cervical edema is massive and extends well off the boundaries of the image (the entire portion of the image ventral to the mandible represents cervical swelling, which is where the asterisk has been placed).

Table 3: This table is not clearly arranged. SEs should appear next to the means they refer to. Please format this table like Table 2.

Corrected, as suggested.

Heading: “Mean (±SD)”, did you mean SE? For in L. 207 you state that all values are reported as mean±SE (Standard error not Standard deviation). In addition, the abbreviation for “subdural” is missing.

Corrected, as suggested.

Footer: Change “p<0.05” to “P<0.05” for style consistency. Change “Methods” to “methods”.

Corrected, as suggested.

L. 358:   Change “respectfully” to “respectively”

Corrected, as suggested.

L. 444:   You mention the positive aspects of such a device, which are unchallenged. However, could you also think of situations in which farmers may kill a rabbit overhasty with such a device, for instance in cases in which separation/veterinary treatment would have been feasible?

From a welfare perspective, we are not concerned with overzealous culling. Under recognition of animals that need to be culled is the real problem on farm. Meat rabbits are generally not treated individually and will die quickly once they start to decline, thus must be dispatched immediately once they are identified.

L. 450:   Besides the esthetically unpleasant aspect of external bleeding, you should also mention the hygienic risks here.

Corrected, as suggested. Biosecurity is compared between the 3 methods in more detail 2 paragraphs down when discussing operator safety.

L. 462:   Please provide a reference for this learning curve.

Added reference, as suggested.

L. 463:   The verb is missing in “Our study suggests that the NPCB device 463 minimal training for proficiency.”.

Corrected, as suggested.

L. 480:   “and”.

Corrected, as suggested.

L. 481:   Do you have any recommendations/suggestions for killing rabbits outside the tested weight range?

We did evaluate multiple methods on rabbits under 150 grams including the ones discussed in this paper and are working on a paper related to this research. We are recommending decapitation for pre-weaned kits at or under 10 d.o..

L. 483:   Although you explain possible welfare implications of an esthetic euthanasia method (L. 444) I would not overrate this aspect and therefore would not mention it in the conclusion. From the point of animal suffering, esthetics is completely irrelevant.

We respectfully disagree. Esthetics is an important but indirect welfare concern if it is the reason a producer is choosing to not intervene and is electing to let an animal die on its own. This is also emphasized in the AVMA Guidelines.

Round 2

Reviewer 1 Report

The authors have kindly and thoroughly addressed previous comments. The resulting paper is much clearer and demonstrates the scale and technical difficulties of working in this area in commercially relevant sites. 

No further comments.

Reviewer 2 Report

Dear authors,

I am happy to see that you have agreed to most of my comments from the first revision round. I think the manuscript has improved significantly. I therefore recommend publication in Animals.